# Visiting Urban Green Space and Orientation to Nature Is Associated with Better Wellbeing during COVID-19

**DOI:** 10.3390/ijerph20043559

**Published:** 2023-02-17

**Authors:** Brenda B. Lin, Chia-chen Chang, Erik Andersson, Thomas Astell-Burt, John Gardner, Xiaoqi Feng

**Affiliations:** 1CSIRO Land & Water, GPO Box 2583, Brisbane, QLD 4001, Australia; 2Department of Evolution and Ecology, University of California, Davis, CA 94720, USA; 3Stockholm Resilience Centre, Stockholm University, 114 19 Stockholm, Sweden; 4Ecosystems and Environment Research Program, University of Helsinki, 00100 Helsinki, Finland; 5Research Unit for Environmental Sciences and Management, North-West University, Potchefstroom 2531, South Africa; 6School of Health and Society, Faculty of Arts, Humanities and Social Sciences, University of Wollongong, Wollongong, NSW 2522, Australia; 7Population Wellbeing and Environment Research Lab (PowerLab), Sydney, NSW 2000, Australia; 8School of Population Health, Faculty of Medicine and Health, University of New South Wales, Sydney, NSW 2052, Australia; 9The George Institute for Global Health, Sydney, NSW 2042, Australia

**Keywords:** urban transitions, pandemic, affinity to nature, affordances, urban design

## Abstract

The COVID-19 pandemic has severely challenged mental health and wellbeing. However, research has consistently reinforced the value of spending time in green space for better health and wellbeing outcomes. Factors such as an individual’s nature orientation, used to describe one’s affinity to nature, may influence an individual’s green space visitation behaviour, and thus influence the wellbeing benefits gained. An online survey in Brisbane and Sydney, Australia (n = 2084), deployed during the COVID-19 pandemic (April 2021), explores if nature experiences and nature orientation are positively associated with personal wellbeing and if increased amounts of nature experiences are associated with improvement in wellbeing in the first year of the COVID-19 pandemic. We found that both yard and public green space visitation, as well as nature orientation scores, were correlated with high personal wellbeing scores, and individuals who spent more time in green space compared to the previous year also experienced a positive change in their health and wellbeing. Consistently, people with stronger nature orientations are also more likely to experience positive change. We also found that age was positively correlated to a perceived improvement in wellbeing over the year, and income was negatively correlated with a decreased change in wellbeing over the year, supporting other COVID-19 research that has shown that the effects of COVID-19 lifestyle changes were structurally unequal, with financially more established individuals experiencing better wellbeing. Such results highlight that spending time in nature and having high nature orientation are important for gaining those important health and wellbeing benefits and may provide a buffer for wellbeing during stressful periods of life that go beyond sociodemographic factors.

## 1. Introduction

As society moves into the third year of the COVID-19 pandemic, it is becoming increasingly apparent that COVID-19 has impacted people’s lives beyond just physical health to issues of economic and social disparities that impact on people’s mental health and wellbeing [1,2]. Issues such as food insecurities, limited access to safe space, and the erosion of social support systems can all impact the wellbeing of individuals [3,4,5]. The sudden and rapid adjustment to the way communities live, work, and interact with each other and the world around us can also lead to unanticipated impacts on health and wellbeing [6].

Evidence from environmental and health research has consistently reinforced the value of spending time in green space for better physical health and mental health [7,8,9]. This has been found to be consistent across cultures and age groups [10,11,12]. Because of this association with improved health and wellbeing, urban green spaces are increasingly recognised as an intervention to build and restore capacities for better mental, physical, and social health, while also supporting mitigation of multiple contextual harms such as air pollution, heat, and noise [13]. Research in parks and gardens during the COVID-19 pandemic has also demonstrated that during the pandemic, access to urban green spaces provided solace and respite, as well as the chance for physical exercise [14,15,16,17]. Gardening during the pandemic was shown to relieve stress and provide social benefits [18,19], and perceived access to urban green spaces during and after the first COVID-19 peak was associated with better health and wellbeing [20]. Even for those who were unable to leave the house, views of green space from windows increased levels of life satisfaction and happiness, while reducing levels of depression, anxiety, and loneliness [21].

Another line of research is rapidly consolidating evidence to show that people with high levels of nature orientation spend more time in green spaces, and therefore gain the health and wellbeing benefits from green space [22,23,24,25]. Nature orientation alone, after controlling for nature experiences, has also been found to be associated with happiness and other subjective wellbeing [22]. Nature orientation aims to quantify an individual’s innate emotional affinity with nature and measure how they assessed their relationship to nature [26]. Research has touched on the implications of connectedness to nature for psychological well-being in more positive emotions and better life satisfaction [26,27]. These studies show that individuals with high levels of nature affinity spend more time in green spaces and thus gain physical health, mental health, and other well-being benefits such as increased social cohesion.

Thus, the COVID-19 pandemic, and the associated pressures and stresses that were indirectly created through rapid lifestyle changes, presents a unique opportunity to better understand how nature orientation and nature experiences may impact on urban residents’ wellbeing during this time. Although studies have shown that urban dwellers sought out green space during restricted periods, this was not equivalent across the population [28,29]. While some research showed that elderly and female members of the community avoided crowded green spaces, research also highlighted that socially disadvantaged groups had more limited access and opportunity to use green space because of other stressors put on their lives (greater work commitment, home schooling, immobility, and lack of access to support services) [15,28].

In this study, we aim to understand if nature experiences and nature orientation were generally positively associated with personal wellbeing and if increased amounts of nature experiences were associated with improvements in personal wellbeing over this initial year of the COVID-19 pandemic. Through an online survey deployed across Brisbane and Sydney, Australia (n = 1050 and n = 1034 respectively), in April 2021, we test if a person’s nature orientation or nature experiences were associated with an individual’s personal wellbeing during this difficult time. Communities living in both cities had experienced lockdowns at the beginning of the COVID-19 pandemic (late-March 2020), in which individuals were only permitted to leave their places of residence for essential needs, only limited gatherings were permitted and non-essential businesses were restricted in their operations [30]. Both cities then experienced sporadic lockdowns during the intervening time between the initial lockdown and the time of the survey in order to control COVID-19 spread [31,32]. April 2021 represented a time period approximately a year after COVID-19’s first lockdowns and restrictions but was also a period of time when neither city was in lockdown, and city dwellers had experienced relatively easy movement in and around the city. Importantly, this ease in movement restriction allowed people to spend time outdoors relatively freely as compared to one year before. The Australian Institute of Health and Welfare reported that healthcare data revealed heightened levels of psychological distress over time, with a rise in the use of mental health services through 2021, and continuing into 2022 [33].

Although restrictions of movement had been relaxed during the time of the survey, it is important to note that citizens were still living with a larger set of stressful conditions and unfavourable changes during that preceding year, with quarantine and home-schooling becoming a sporadic occurrence, loss of employment or working from home, as well as social isolation from loved ones due to domestic and international border restrictions [33,34]. We hypothesize that a person may have better psychological wellbeing if they visit green spaces more often during periods of difficulty, as we have seen during the COVID-19 pandemic, and a person’s nature orientation may also positively correlate with psychological wellbeing. In addition, we also hypothesise that individuals who increase their green space visit over the year may experience improvements in subjective health and wellbeing.

## 2. Methods

### 2.1. Location of Study

This study was performed in two of Australia’s largest cities, Sydney and Brisbane. Both cities are coastal cities located on the east coast of Australia, approximately 900 km apart.

Sydney is the capital of New South Wales and the most densely populated city in Australia. It has a population of approximately 5.35 million over an area of 12,000 km^2^ [35]. The Greater Sydney area exhibits great variation in the amount of tree cover within the city, with an average of about 20% tree cover for the region. There is currently an aim to increase this cover to 40% [36]. Annual average temperatures range from 13.8 °C (mean minimum) to 21 °C (mean maximum) with an annual average rainfall of 1213.4 mm [37].

Brisbane is the capital of Queensland and the third most populous city in Australia. It has a population of approximately 2.6 million over an area of about 16,000 km^2^ [35]. The city exhibits high overall levels of public green space (>200 m^2^ per person) and tree cover (36%), both of which are spread rather evenly across the socio-economic gradient [38]. Annual average temperatures range from 15.7 °C (mean minimum) to 25.5 °C (mean maximum) with an annual average rainfall of 1148.8 mm [39].

### 2.2. Survey Information

An online survey was conducted between 15 April and 15 May 2021 for Brisbane and Sydney residents, asking about their nature experiences and nature orientation. This research was conducted in accordance with approved guidelines, and all protocols were received under Institutional Human Research Ethics Approval (CSIRO Human Research Ethics Review Board, Project 144/20). Informed consent was obtained from all respondents.

The survey was delivered by an online data collection company, the Online Research Unit, a general market and data analysis company well-established in Australia, to run a survey panel through their existing research databases of potential respondents in each city. The time period was chosen as it was during a time when seasonal temperatures would not affect participation in going to nature spaces. Demographic parameters were provided to the company to ensure that sampling occurred across gender, education, and income variables. A minimum high-quality sampling number was requested from the survey company (n = 1000) for each city, with a total of 1050 respondents captured from Brisbane and 1034 surveys collected from Sydney for a total of 2084 responses.

To test if nature orientation and nature experience were associated with better wellbeing, participants were asked to recall how often they visited their yards (private green space that is directly around their home, sometimes called gardens) or public green space (frequency of yard visits, frequency of green space visits), how long they spent in their yards or public green space last week (duration of yard visits, duration of green space visits), and whether they spent more or less time in their yard or public green space over the year (change in yard visits over the year, change in green space visits over the year). We also quantified a participant’s nature orientation score and personal wellbeing, and how they perceived their health and wellbeing might have changed over the year. The descriptive statistics of these two cities are shown in Appendix A.

#### 2.2.1. Socio-Demographic Information and Nature Connection

For this study, a large range of questions were asked regarding a survey participant’s self-reported socio-demographic information, their use of green space, nature orientation, personal well-being, and change in perceived health and wellbeing across the year. Socio-demographic questions included information on age, gender, education level, and income level. Survey participants were asked to complete the Nature Relatedness Scale (referred to as ‘NR’ here) to assess their level of nature orientation [40]. This scale requires participants to complete a series of questions that assess the affective, cognitive, and experiential relationship individuals have with the natural world across 21 statements. These responses were then scored and calculated according to the process presented in Nisbet et al. [40]. A higher average score indicates a stronger connection with nature. The scale has been demonstrated to differentiate between known groups of nature enthusiasts, as well as those who do and do not self-identify as environmentalists. It also correlates with environmental attitudes and self-reported behaviour and appears to be relatively stable over time and across situations [40]. This NR Scale has been validated and tested across different communities [41,42,43].

#### 2.2.2. Current Green Space Visitations

Current green space visits include frequency and duration of yard visits and frequency and duration of public green space visits and have been used in previous surveys [24,44,45,46,47].

Frequency of yard visits*:* Participants were asked to recall how often they usually spend more than 10 min in their own yard or on their deck. The frequency was selected from the following categories: I don’t have a yard or deck (=0), never (=0), less than once a month (=1), 2–3 times a month (=2), once a week (=3), 2–3 days a week (=4), 4–5 days a week (=5), and 6–7 days a week (=6). Participants chose “I don’t have a yard or deck were considered as zero frequency in yard visit.

Duration of yard visits last week: Participants were also asked to think about the last week, how much time in total they spent in their own yard or on their deck. The duration was selected from the following categories: No time (=0), 1–30 min (=1), 31 min to 1 h (=2), 1–3 h (=3), 3–5 h (=4), 5–7 h (=5), 7–9 h (=6), more than 9 h (=7). Similar to the frequency of yard visits, participants chose “I don’t have a yard or deck were considered as zero duration of yard visit last week.

Frequency of public green space visits*:* Participants were asked to recall how often they usually visited or passed through outdoor green spaces for any reasons. The frequency was selected from the following categories: never (=0), once a year (=1), once every three months (=2), once a month (=3), 2–3 times a month (=4), once a week (=5), 2–3 days a week (=6), 3–5 days a week (=7), and 6–7 days a week (=8).

Duration of public green space visits: Participants were asked to recall over the last week what outdoor green spaces they visited or travelled through and to estimate the total time they spent there in hours. Participants who reported spending more than 168 h or less than 0 h in public green spaces were considered as error (“NA”). Participants who did not visit any public green spaces were considered as zero duration of public green space visits last week.

#### 2.2.3. Change in Green Space Visitation over the Previous Year—In Public Green Space and Private Yards

In order to gauge how green space use had potentially changed over the past year (essentially during the first year of COVID-19 impacts on lifestyle), we asked respondents two sets of questions:Compared to this time last year, are you spending a different amount of time in your yard?Compared to this time last year, are you spending a different amount of time in outdoor green spaces? This includes, for example, beaches, bushland, playgrounds or picnic areas, dog off-leash areas, national parks.

Responses were categorical and classed as: much less, less, same, more, much more. Within the analysis, change in use —coded as −2, −1, 0, 1, 2, respectively. Respondents who selected that they did not have a yard were coded as not applicable (na) within the dataset.

#### 2.2.4. Personal Wellbeing Measures

Personal wellbeing was assessed using the Personal Wellbeing Index developed by the Australian Centre on Quality of Life (ACQOL, www.acqol.com.au, accessed on 16 February 2023). This consortium research group examines quality of life as both an objective and subjective dimension, which comprises several domains. These domains together define the total construct [48]. This scale contains seven items of satisfaction, each one corresponding to a specific quality of life domain, which includes: standard of living, health, achievement in life, relationships, safety, community-connectedness, and future security. The PWI has been validated across user groups and is used in cross-cultural settings [49].

#### 2.2.5. Change in Personal Wellbeing over the Previous Year

In addition to the Personal Wellbeing set of seven questions, a following question was asked to gauge an individual’s self-assessment of their change in health or well-being over the last year.

Compared to this time last year, how has your health or well-being changed?

Responses were categorical and classed as: much worse, worse, same, better, much better. Within the analysis, change in use was coded as −2, −1, 0, 1, 2, respectively. As relatively few people answered that their wellbeing was much worse (−2) or much better (2), we recoded the data as having positive change (including better and much better) or having negative change (including worse and much worse).

### 2.3. Analysis

The data analyses were performed using R v4.1.2 (accessed on 1 November 2022).

#### 2.3.1. Current Personal Wellbeing Index

To model the relationship between current nature experiences (duration and frequency of yard visits, and duration and frequency of public green space visits) and a person’s personal wellbeing, we used a linear regression model with personal wellbeing index as the response variable. Explanatory variables were duration and frequency of yard visits, and duration and frequency of public green space visits. We also included a person’s nature relatedness. The demographic factors included in the model were age, gender, income, and city. We did not detect any multi-collinearity (R package car with vif function, 3 as a cut off), and the model fulfilled homoscedasticity and normality assumptions. We also ran a model without individuals not having a yard (na) as a sensitivity analysis, and the result was consistent (Appendix A).

#### 2.3.2. Change in Health or Wellbeing over a Year

To investigate whether/how change in green space visits may influence the change in health or wellbeing, we ran two separate generalized linear models with binomial error structure: (1) the likelihood to have positive change and (2) the likelihood to have negative change.

In the first model, we used the dataset where participants reported to have no change or positive change. The response variable was the no/positive change, which was coded as 0 or 1. Explanatory variables were the change in yard visits across a year and change in green space visits across a year. As current nature experiences may also influence the perception in the change in wellbeing, we also included time spent in the yard last week, frequency of yard visit over 10 min this year, frequency of green space visits this year, time spent in public green space last week, and nature relatedness. The demographic factors included in the model were age, gender, income, and city (Sydney or Brisbane). In the second model, we used the dataset where participants reported to have no change or negative change. The response variable is the no/negative change, which was coded as 0 or 1. The explanatory variables and covariates were the same as the first model. It is important to note that we did not run an additional model excluding individuals without a yard (as we did for the PWI model), because the change in yard use is not applicable for individuals without a yard.

## 3. Results

### 3.1. Associations between Current Green Space and Yard Visitation and Current Personal Wellbeing

We found that people who visited their yards or urban green space more often had higher self-reported personal wellbeing, after adjusting for the confounding effects of age, gender, income, and city (Table 1, Figure 1). We also examined the association between nature orientation and wellbeing and found that people who had a stronger nature orientation score reported better personal wellbeing (Table 1). Demographic factors were also included in the analysis to account for potential confounders, and age and income were associated with positively personal wellbeing scores (Table 1).

### 3.2. Green Space Use over the Year and Reported Change in Wellbeing

We analysed the change in status wellbeing and health to better understand how changes in green space use were related to an increased likelihood of reporting positive or negative change in health and wellbeing over the year. We found that individuals who spent more time in green space compared to the previous year were more likely to experience a positive change in their health and wellbeing, and they were also less likely to experience a negative change in their health and wellbeing (Table 2, Figure 2). Individuals who visited their yard more often compared to the previous year (Figure 2) or individuals with a higher nature orientation experienced a positive change in wellbeing (Table 2). Age was negatively correlated with positive and negative change in wellbeing over the year, meaning older individuals were less likely to report a change in wellbeing; while income was negatively correlated with a negative change in wellbeing over the year, meaning it was less likely for people with a higher income to have a negative change in wellbeing (Table 2).

## 4. Discussion

This study supports current research that there is a distinct effect of green space visitation for improved health and wellbeing outcomes. During the COVID-19 pandemic, individuals who visited a green space or private yard more often, or had a stronger nature orientation, reported a higher level of wellbeing. Additionally, over the course of the year from around April 2020 to April 2021, individuals who visited public green spaces more reported themselves as having a positive change in wellbeing over the course of the year, more so than in yard visits, although there was also a positive change. Consistently, individuals who had visited green spaces more had a lower likelihood of saying that they experienced a negative change in wellbeing. Interestingly, people with a stronger connectedness with nature also experienced a positive change in wellbeing.

Based on these correlations, public green spaces appear to serve a critical role in benefiting a person’s wellbeing score during this difficult year of the COVID-19 pandemic. Many cities in many countries globally would have experienced similar lockdowns and mobility restrictions that impacted on health and wellbeing [50,51]. As was seen in pre-pandemic studies, urban green spaces provided many health and wellbeing benefits for city dwellers [8,9]. This study further demonstrates that urban green spaces not only continue to benefit the wellbeing of individuals and our urban communities but also buffer negative impacts during the difficult periods, as increased frequency in public green space visits are associated with a lower likelihood of negative change in wellbeing over the year. This suggests green spaces are critical, if not essential, for the health and wellbeing of city dwellers.

In addition to visiting green spaces, we also see a positive correlation between nature orientation and wellbeing during the COVID-19 pandemic. There are two possible mechanisms. First, people with higher nature orientation visit green spaces more often in general, and they may have already been using green spaces at a higher rate before the lockdowns [44]. We propose that the green space visits may have a long-lasting effect on wellbeing and already built capacities for psychological resilience from the nature experienced before the onset of COVID-19. Second, previous barriers to accessing and engaging with nature may have been reduced during the pandemic, including more time at home to spend in nature or access local green spaces [52]. With that being said, people with a higher nature orientation score may increase green space visitations during the pandemic [16,28,53] or spend more time at home with a nature view through windows that may be missing in the workplace [21,54].

We found that income is associated with personal wellbeing and likelihood to have a negative change in wellbeing. This result is consistent with previous studies. Studies from adult populations have shown that adult cohorts from lower socio-economic status households spent less time outside compared with adults from higher socio-economic status households [16,55]. This may also be related to having a greater income that assists in buffering the stresses of COVID-19, such as white-collar jobs and the ability to work from home, the ability to pay for help with home schooling, and greater access to nature spaces in more economically advantages areas [56,57]. In Sydney, higher infection numbers were located in the councils with the lowest household income, due to challenges with living in a ‘bubble’ or staying socially distanced with work and family; these areas also faced harsher and longer lockdown restrictions for these reasons [58].

Additionally, older individuals were less likely to have experienced a change in wellbeing compared to last year (positive or negative). This may be related to the use of public vs. private green spaces. Older individuals, who were worried about getting sick and contracting COVID-19, may have felt uncomfortable going into public green spaces due to the increased exposure to other people; however, they may in turn use private green space more often. In this case, there is a need to consider how private yards may help deliver nature interactions when public green spaces are not seen as ‘safe’ to visit by some people. A study from Brisbane in 2020 found that older people were less likely to use public green space during lockdown periods, and people with backyards or private green space were more likely to use these spaces when there were restrictions in place [29]. Another study that aimed to help older individuals experience diverse outdoor spaces during the COVID-19 pandemic found that outdoor engagement led to increased perceived social, mental, and physical well-being [59].

There are some limitations in this study. First, this is a correlative study; therefore, we can only speculate on the directionality of the causality based on other literature. We could not exclude the possibility that, for example, people with better health and wellbeing status chose to visit greenspace more often than people with worse health or wellbeing status. Second, we were also unable to exclude the potential confounding factors that were not captured in the survey. For example, individuals who visit greenspace more often may have more control over their time, which may lead to the better wellbeing. Third, older individuals were less likely to report a change in wellbeing over the year. Though it may be linked with green space visits, it may also be because the lockdown and mobility restriction has less impact on them as compared to working adults or students. Further research should aim to disentangle how these individual demographic factors may interact with psychological factors, such as orientation to nature, to better understand the way different populations use and engage green spaces.

## 5. Conclusions

This combination of results indicates that nature relatedness, coupled with green space visitation, were able to help mitigate the negative impacts of COVID-19 stressors on human wellbeing in these two cities and allow individuals to experience a positive change in wellbeing. As COVID-19 continues to impact on the decision-making and behaviour around green space use, policies need to be considered to help individuals access green space safely in new situations, especially if we are to move toward a more inclusive and sustainable future where everyone has access to the mental health and wellbeing benefits of green spaces [60,61]. This paper shows that nature orientation continues to be an important factor that influences green space use and wellbeing, and further research is required to understand how individuals are using green space and interacting with nature to gain these wellbeing benefits.

## Figures and Tables

**Figure 1 ijerph-20-03559-f001:**
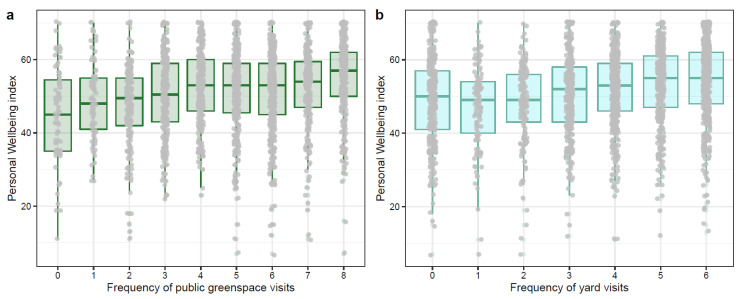
Differences in personal wellbeing based on the frequency of public green space and yard visit: People who visit public green space more often (**a**) or visit the yard more often (**b**) were more likely to have better personal wellbeing. Grey points are raw data points. In (**a**), 0 = never, 1 = once a year, 2 = once every three months, 3 = once a month, 4 = 2–3 times a month, 5 = once a week, 6 = 2–3 days a week, 7 = 3–5 days a week, and 8 = 6–7 days a week. In (**b**), 0 = without access to yard or never, 1 = less than once a month, 2 = 2–3 times a month, 3 = once a week, 4 = 2–3 days a week, 5 = 4–5 days a week, and 6 = 6–7 days a week. Additional analysis was run excluding individuals without access to a yard (Appendix A).

**Figure 2 ijerph-20-03559-f002:**
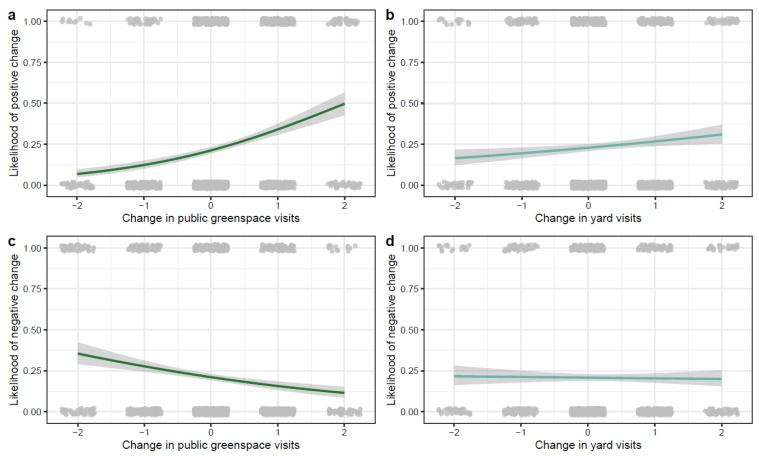
Likelihoods of change based on green space and yard visits: People who increased their (**a**) public green space and (**b**) yard visits over the year were more likely to have positive change in health and wellbeing. Consistently, people who increased their public green space visits compared to last year were less likely to have negative change in health and wellbeing (**c**), but this pattern was not seen in changes to yard visits (**d**) −2 = much less, −1 = less, 0 = same, 1 = more, 2 = much more.

**Table 1 ijerph-20-03559-t001:** The associations between personal wellbeing and nature experiences and nature orientation after controlling for demographic factors.

	Estimate	SE	*p*
(Intercept)	34.873	1.835	<0.001
**Frequency of yard visit**	**0.708**	**0.187**	**<0.001**
Duration of yard visit	−0.225	0.185	0.223
**Frequency of public green space visit**	**0.483**	**0.127**	**<0.001**
Duration of public green space visit	0.060	0.082	0.465
**Nature relatedness**	**1.194**	**0.451**	**0.008**
**Age**	**0.485**	**0.076**	**<0.001**
Gender (male)	0.022	0.510	0.966
**Income**	**0.684**	**0.093**	**<0.001**
City (Sydney)	−0.864	0.496	0.082

Bold is to show what variables are significant.

**Table 2 ijerph-20-03559-t002:** Change in nature experience and change in health and wellbeing (positive and negative) over the first year of the COVID-19 pandemic, after controlling for demographic factors.

	Positive Change			Negative Change		
	Estimate	SE	*p*	Estimate	SE	*p*
(Intercept)	−3.471	0.538	<0.001	−1.697	0.523	0.001
Chang in yard visits	0.021	0.086	0.803	−0.001	0.091	0.990
**Change in green space visits**	**0.548**	**0.099**	**<0.001**	**−0.293**	**0.097**	**0.002**
Duration of yard visit	−0.098	0.049	0.047	0.030	0.050	0.553
**Frequency of yard visit**	**0.138**	**0.059**	**0.019**	0.005	0.060	0.939
Frequency of public green space visit	0.044	0.039	0.257	0.016	0.038	0.678
Duration of public green space visit	0.018	0.022	0.398	0.008	0.024	0.733
**Nature relatedness**	**0.642**	**0.131**	**<0.001**	0.230	0.131	0.079
**Age**	**−0.172**	**0.023**	**<0.001**	**−0.043**	**0.022**	**0.048**
Gender (male)	0.209	0.143	0.145	0.003	0.147	0.985
**Income**	0.030	0.027	0.266	**−0.065**	**0.026**	**0.012**
City (Sydney)	−0.118	0.139	0.399	0.129	0.143	0.367

Bold is to show what variables are significant.

## Data Availability

All data generated and analysed as part of this study will be made available upon request and in an online repository once the collective set of papers have been published. Data codes for the analysis presented in this paper are available at this figshare link: https://doi.org/10.6084/m9.figshare.22056869.v1 (accessed on 16 February 2023).

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
