# Peer review of "Visiting Urban Green Space and Orientation to Nature Is Associated with Better Wellbeing during COVID-19"

_ijerph, 2023, doi:10.3390/ijerph20043559_

Round 1

Reviewer 1 Report

Overall, the article is well written and the topic is relevant.

The following change in the title is suggested: "Visitation to urban green spaces and orientation to nature associated with better wellbeing during Covid-19".

Author Response

Thank you for your review. We have adjusted the title to reflect the suggested change.

Reviewer 2 Report

The manuscript presents results from a survey in Australia regarding wellbeing during the Covid pandemic. Nature affinity and greenspace are examined. The findings show that using greenspace during the pandemic was associated with improved wellbeing. Overall the paper is well written and nicely assembled.

A few overarching comments to point out:

1. As is stated in the limitations, it would be difficult to say if using greenspace resulted in better wellbeing or if better wellbeing resulted in a person using greenspace more. While causality is avoided, like many studies we are kind of interested in causes.

2. This manuscript does fall foul of the "Table 2 fallacy", though it is table 1 here. There is a good reference for this - Westreich, D., & Greenland, S. (2013). The table 2 fallacy: presenting and interpreting confounder and modifier coefficients. American Journal of Epidemiology, 177(4), 292–298. Essentially it is the interpretation of all coefficients in the model output (line 286 for example, but also in the results). We all do this but I think the winds are changing in the research environment.

3. If the analysis was done in R it would be nice to make the code available somewhere like GitHub. It is OK if the data is not available until later, but sharing the code means others can learn from your efforts.

The above are just comments and don't really require any response.

Some things I would like to see addressed are:

1. The term "yard" doesn't show up until line 158. I would define exactly what you mean by this. Sometimes the same thing is called a "garden" depending on what culture you are in.

2. The paragraph starting at line 80 is a little repetitive between the first sentence and the last. Maybe that could be edited some.

3. Having the sensitivity analysis is good (line 257). I'm not completely sure that not having a yard should mean 0 yard visits, but I'm OK with leaving that in (like line 193). 

4. You may know better for the 2 cities, but could there be some underlying reason a person does not have a yard? It would seem that greater socioeconomic status would correlate with having a yard. If you think there is a reason then perhaps this could be included in the methods or limitations.

5. The survey question shown on line 239 kind of has 2 concepts captured in it - health and wellbeing. It seems that knowing which the respondent was responding to would be unclear. I suppose there isn't much you can do about it now though.

6. On line 277, so were these people excluded from the second model?

7. The last sentence of the manuscript seems a little out of place. The piece doesn't really address issues of disparities in wealth or access. Income is included in the modelling but that alone doesn't make the study about differences. I would just drop that sentence and add something to the previous sentence about how this study points to the value of considering who does and does not have access to greenspaces for the benefits you found.

8. I would add something to the limitations like a "further research...". Maybe you can point to what you think should be investigated next based on your work (and its limitations).

Overall it is a good contribution to the growing greenspace literature. The data source is interesting and provides work for others to build on.

Author Response

The manuscript presents results from a survey in Australia regarding wellbeing during the Covid pandemic. Nature affinity and greenspace are examined. The findings show that using greenspace during the pandemic was associated with improved wellbeing. Overall the paper is well written and nicely assembled.

Thank you for your comments and taking the time to review this paper.

A few overarching comments to point out:

  1. As is stated in the limitations, it would be difficult to say if using greenspace resulted in better wellbeing or if better wellbeing resulted in a person using greenspace more. While causality is avoided, like many studies we are kind of interested in causes.
  2. This manuscript does fall foul of the "Table 2 fallacy", though it is table 1 here. There is a good reference for this - Westreich, D., & Greenland, S. (2013). The table 2 fallacy: presenting and interpreting confounder and modifier coefficients. American Journal of Epidemiology, 177(4), 292–298. Essentially it is the interpretation of all coefficients in the model output (line 286 for example, but also in the results). We all do this but I think the winds are changing in the research environment.
  3. If the analysis was done in R it would be nice to make the code available somewhere like GitHub. It is OK if the data is not available until later, but sharing the code means others can learn from your efforts.

The above are just comments and don't really require any response.

Thank you for these comments. Although we recognise that you don’t expect a response, we believe that the review process is a great way to have a conversation about these topics. 

  1. Yes, we agree that causality is of huge interest, and we are aiming for future research to test hypotheses of causality.
  2. Although we understand the concern for Table 1, we felt that because there was only one table, it would be helpful for readers to see all the variables, especially because socio-demographic variables of play a role in green space visitation.
  3. We can supply the code as a figshare link. We have included the link to the Data Availability statement. We will make the data available at a later point in an online repository, but are happy for enquiries to come to us for now. We will update the manuscript to state these changes.

Some things I would like to see addressed are:

  1. The term "yard" doesn't show up until line 158. I would define exactly what you mean by this. Sometimes the same thing is called a "garden" depending on what culture you are in.

We have added an explanation directly after the term on line 158 stating “(private green space that is directly around their home, sometimes called gardens)”.

  1. The paragraph starting at line 80 is a little repetitive between the first sentence and the last. Maybe that could be edited some.

We have moved the last sentence to the next paragraph as it is introducing the study and the rationale of the presented work. (line 94-97)

  1. Having the sensitivity analysis is good (line 257). I'm not completely sure that not having a yard should mean 0 yard visits, but I'm OK with leaving that in (like line 193). 

We understand the reviewer’s concern about this decision.  We had considered a number of ways to try to address this issue. However, because we do not have more specific information on if they visit other people’s yards or have a shared green space, we assume that respondents who say they don’t have yards do not have immediate-available private green space that they can visit.

  1. You may know better for the 2 cities, but could there be some underlying reason a person does not have a yard? It would seem that greater socioeconomic status would correlate with having a yard. If you think there is a reason then perhaps this could be included in the methods or limitations.

Many of the people in these two cities who do not have a yard live in apartments. However, it is difficult to associate a socioeconomic trend to this pattern in these two cities.  Many individuals who live in apartments choose to do so out of convenience to inner city jobs are enjoy the amenities of more built-up areas of the city.  These are both owners and renters and cover wealthy and less wealthy areas.

  1. The survey question shown on line 239 kind of has 2 concepts captured in it - health and wellbeing. It seems that knowing which the respondent was responding to would be unclear. I suppose there isn't much you can do about it now though.

You are correct that these two concepts are captured within the question.  Our aim was to get a general understanding of whether respondents felt there was improvement or if their general health and wellbeing had decreased. 

  1. On line 277, so were these people excluded from the second model?

Yes, this is correct.  We excluded these individuals from the model to ensure that the inclusion of these individuals in the first model was consistent even when these individuals were excluded.

  1. The last sentence of the manuscript seems a little out of place. The piece doesn't really address issues of disparities in wealth or access. Income is included in the modelling but that alone doesn't make the study about differences. I would just drop that sentence and add something to the previous sentence about how this study points to the value of considering who does and does not have access to greenspaces for the benefits you found.

We have erased this sentence and added to the previous sentence: “…, and further research is required to understand how individuals are using green space and interacting with nature to gain these wellbeing benefits.” (line 406-407)

  1. I would add something to the limitations like a "further research...". Maybe you can point to what you think should be investigated next based on your work (and its limitations).

We have added to a sentence “Further research should aim to disentangle how these individual demographic factors may interact with psychological factors, such as orientation to nature, to better understand the way different populations use and engage green spaces.” (line 410-411)

Overall it is a good contribution to the growing greenspace literature. The data source is interesting and provides work for others to build on.

Thank you for all your suggestions.

Reviewer 3 Report

There have been a number of these studies that have come to similar conclusions for you.  For example,Significant global health challenges, such as the negative effects of physical and psychological symptoms, are being confronted in the 21st century (Beaglehole et al., 2012; Danielsson et al., 2012). For example, COVID-19 has increased both disease mortality and mental health problems (Greenberg, 2020; Rajkumar, 2020), and psychological problems are difficult to eliminate in the short term (Brooks et al., 2020). These factors, combined with population growth, rapid urbanization and climate change, indicate that prevention approaches must be reconsidered (Das and Horton, 2012; Watts et al., 2015). With the increase in the urban population, the relationship between humans and nature is being increasing considered during urban planning (Bush and Doyon, 2019; McEwan et al., 2020). Many studies have shown that short-term stays in natural environments can improve mental health (Beil and Hanes, 2013; Tyrväinen et al., 2014; M. P. White et al., 2013) and reduce stress (Hartig et al., 2003). Therefore, restorative environments are receiving increasing attention in urban environment research (Frumkin, 2001).Restorative environments are typically natural environments with green and blue space (Labib et al., 2020;  Mears et al., 2019;  Voelker et al., 2016).

In addition, Zhao et al. studied the impact of landscape characteristics and soundscapes on the restorative quality of urban green space (Zhao et al., 2018).(Zhao, J., Xu, W., Ye, L. 2018. Effects of auditory-visual combinations on perceived restorative potential of urban green space. Applied Acoustics, 141, 169-177. doi:https://doi.org/10.1016/j.apacoust.2018.07.001)

And (Deng, L., Luo, H., Ma, J., Huang, Z., Sun, L.-X., Jiang, M.-Y., et al. 2020. Effects of integration between visual stimuli and auditory stimuli on restorative potential and aesthetic preference in urban green spaces. Urban Forestry & Urban Greening, 53, 126702. doi:https://doi.org/10.1016/j.ufug.2020.126702)

Therefore,you need talk about the necessity and specificity of your research?

Author Response

Thank you for your comments.  We agree that there has been very good research in the past showing that green spaces are important for health and wellbeing.  We acknowledge that in paper in paragraph 2 (lines 49-66) with many citations, as you suggest. 

However, the goal of this manuscript was to highlight the importance of green space during a time where mental health and wellbeing was especially challenged.  In this case, we note that Covid-19 impacted many people with movement restrictions, job losses, and loss of daily social activity impacting mental health and wellbeing.  We aim to study this period in time (the first year of the Covid-19 pandemic) to better understand the role of private and public green space for individuals and their mental health and wellbeing during a very difficult period. Because Covid-19 was handled in such different ways around the world, it is important to cover and report on instances in different contexts. Having studies from around the world will allow us to understand the diversity of experiences that occurred under Covid-19 and provide evidence of this vase range of experiences in the future. 

We introduce this concept in the first paragraph to emphasise the scope of the paper (line 40-48) and reemphasise the potential of Covid-19 to challenge mental health in paragraph 4 of the introduction (line 80-93).

Round 2

Reviewer 3 Report

Accept in present form